# Progress & Prospect of Enzyme-Mediated Structured Phospholipids Preparation

Yuhan Li [1], Lingmei Dai [1], Dehua Liu [1,2] and Wei Du [1,*]

[1] Key Laboratory for Industrial Biocatalysis, Ministry of Education, Department of Chemical Engineering, Tsinghua University, Beijing 100084, China; yh-li21@mails.tsinghua.edu.cn (Y.L.); dailm@126.com (L.D.); dhliu@tsinghua.edu.cn (D.L.)

[2] Tsinghua Innovation Center in Dongguan, Dongguang 523808, China

[*] Correspondence: duwei@tsinghua.edu.cn

**Abstract:** In recent years, structured phospholipids (SPLs), which are modified phospholipids (PLs), have attracted more attention due to their great potential for application in the field of pharmacy, food, cosmetics, and health. SPLs not only possess enhanced chemical, physical and nutritional properties, but also present superior bioavailability in comparison with other lipid forms, such as triacylglycerols, which make SPLs become more competitive carriers to increase the absorption of the specific fatty acids in the body. Compared with chemical-mediated SPLs, the process of enzyme-mediated SPLs has the advantages of high product variety, high substrate selectivity, and mild operation conditions. Both lipases and phospholipases can be used in the enzymatic production of SPLs, and the main reaction type contains esterification, acidolysis, and transesterification. During the preparation, reaction medium, acyl migration, water content/activity, substrates and enzymes, and some other parameters have significant effects on the production and purity of the desired PLs products. In this paper, the progress in enzymatic modification of PLs over the last 20 years is reviewed. Reaction types and characteristic parameters are summarized in detail and the parameters affecting acyl migration are first discussed to give the inspiration to optimize the enzyme-mediated SPLs preparation. To expand the application of enzyme-mediated SPLs in the future, the prospect of further study on SPLs is also proposed at the end of the paper.

**Keywords:** catalysis; lipase; phospholipase; phospholipids; structured phospholipids (SPLs)

## 1. Introduction

Phospholipids (PLs) are the main structural and functional components of cell membranes [1], which usually consist of glycerol, fatty acids, a phosphate group, and a polar headgroup (Figure 1). The three carbon atoms of glycerol are the main carbon backbone of the phospholipid molecule. The *sn*–1 and *sn*–2 positions of glycerol are esterified with fatty acids ($R_1$ and $R_2$ in Figure 1), while the phosphoric acid linked to the *sn*–3 position of glycerol can be further esterified with various polar headgroups, such as choline, serine, ethanolamine, glycerol, hydrogen, or inositol, to form phosphatidylcholine (PC), phosphatidyl serine (PS), phosphatidyl ethanolamine (PE), phosphatidylglycerol (PG), phosphatidic acid (PA), or phosphatidylinositol (PI), respectively (Figure 1) [2]. The hydrophilic "head" and hydrophobic "tail" make the PL molecule amphiphilic, hence it can create a lipid bilayer to facilitate the passage of fatty acids (FAs) through the cell membranes and increase the bioavailability of FAs [3,4].

In general, PLs can be isolated and characterized from various sources, containing plant (vegetable oils, especially soybean), animal (egg yolk and cheese whey), and aquatic sources (fish and krill) [2]. The types of natural PLs are limited and the separation of specific PLs is quite difficult for they rarely exist in nature [5]. Both the position and composition of fatty acyl determine the properties of phospholipids, such as antioxidant activity [6], lubricity [7], and stability [8].

**Figure 1.** The molecular structures of phospholipids. The structure usually consists of glycerol, fatty acids (R$_1$ and R$_2$), a phosphate group, and a polar headgroup (X). The phosphoric acid linked to the *sn*–3 position of glycerol can be further esterified with various polar headgroups, such as choline, serine, ethanolamine, glycerol, hydrogen atom, or inositol, to form phosphatidylcholine (PC), phosphatidyl serine (PS), phosphatidyl ethanolamine (PE), phosphatidylglycerol (PG), phosphatidic acid (PA), or phosphatidylinositol (PI), respectively.

To make a wide range of PLs available as well as enhance their performance, the preparation of structured phospholipids (SPLs) has raised attention. SPLs have been used in the food, pharmaceutical, and cosmetics industries and functioned as emulsifiers, stabilizers, and antioxidants [5]. Recently, plenty of research reported that SPLs could be used to transport pharmaceutical molecules, such as betulinic acid [9], usnic acid [10], paeoniflorin [11], anisic (ANISA) and veratric acid (VA) [12]. These drug delivery systems could increase the adsorption as well as enhance the stability of pharmaceutical molecules, thus greatly improving the bioavailability. Besides, some nutrients, such as docosahexaenoic acid (DHA), the main omega-3 polyunsaturated fatty acid in brain tissues crucial to common brain growth and function, can be accumulated and utilized more efficiently in SPLs forms than that in triglycerides (TAG) forms [13,14].

Via the transformation of fatty acids or polar headgroups on the native PLs, structured phospholipids (SPLs) are obtained with enhanced physical or chemical properties, thus having the potential to target specific diseases and metabolic conditions [1]. The methods to prepare SPLs can be divided into two kinds: chemical-mediated and enzyme-mediated. For the chemical-mediated method, chemicals are used to react with the functional groups (such as the carbon—carbon double bond and polar head group) of PLs, thus resulting in chemical change of PLs and creating new kinds of PLs. Zhang, et al. [15,16] have used mesoporous organ sulfonic acid-functionalized SBA-15 catalysts and sulfonated Zn-SBA-15 catalysts to prepare structured phospholipids containing medium-chain fatty acids (MCFA) and short-chain fatty acids (SCFA), respectively. Their results proposed that heterogeneous catalysts with mesopores might be more efficient, economical, and eco-friendly for the production of SPLs with MCFA and SCFA. However, some chemicals that chemical-mediated methods use have poor security and fail to meet the food additive standard of some countries, thus limiting their use in the food and the pharmaceutical industry. Compared with chemical-mediated methods, enzyme-mediated methods can offer considerable advantages, such as high selectivity and mild conditions, leading to the generation of products that cannot easily be obtained by chemical methods [5,17]. Lipases and phospholipase are two main kinds of

enzymes used to catalyze the production of SPLs, and they are widely found in animals (specific cells that synthesize lipases, or other host animals present in the gastrointestinal tract in animals to digest fats and lipids), plants (seeds or grains, fruits, leaves, and some other organisms) and microorganisms (fungi, yeast, and bacteria) [18–21].

In recent years, several reviews on the SPLs have been published. Sun, Chen, Wang and Lin [2] presented a detailed review of the sources, molecular species, and structures of food-derived phospholipids. Wang, et al. [20] summarized developments in the formation of versatile phospholipid assemblies, together with the applications of these assemblies in building artificial tissues and biomedical applications. Ang, Chen, Xiang, Wei and Quek [1] provided information on the preparation method of SPLs, addressing the potential health benefits and the methods used to analyze the SPL profile and metabolism. This review, however, mainly focuses on the progress of enzyme-mediated structured phospholipids preparation, including the types of enzymes that are usually used in the production of SPLs with their corresponding strategies of reactions, and parameters affecting the SPLs production with related mechanisms. The progress in enzymatic modification of PLs over the last 20 years is reviewed. To provide ideas for optimizing the enzyme-mediated SPLs preparation, reaction types and characteristic parameters are summarized in detail and the parameters affecting acyl migration are first discussed in this paper. Finally, the prospect of further study of SPLs is proposed to expand the application of enzyme-mediated SPLs in the future.

## 2. Lipase-Catalyzed Structured Phospholipids

Lipases (E.C.3.1.1.3), also known as triacylglycerol hydrolase, can catalyze reactions containing esterification, hydrolysis, transesterification, acidolysis, and some other important reactions [21]. Depending on the acting site, lipases can be further classified into *sn*–1,3 position-specific enzymes and non-specific enzymes. There are three main strategies: esterification, acidolysis, and transesterification.

### 2.1. Esterification

Esterification is the reverse reaction to hydrolysis and conducts the production of an ester and water from an acid and alcohol [22]. Lipases can catalyze the esterification between *sn*-glycerol-3-phospholipids (GPL) (Figure 2a) or lysophospholipids (LPL) (Figure 2b,c) and free fatty acid (FFA) to produce corresponding SPLs (Figure 2). Wang and her colleagues used immobilized MAS1 lipase to synthesize lysophosphatidylcholine (LPC) enriched with n-3 polyunsaturated fatty acids (n-3 PUFA) [23]. They found that immobilized MAS1 lipase is an efficient biocatalyst for the synthesis of n-3 PUFA-rich LPC by the esterification of *sn*-glycerol-3-phosphatidylcholine (GPC) with n-3 PUFA. Hong, et al. [24] reported that LPC was successfully synthesized by enzyme-catalyzed esterification of GPC with the acid form of conjugated linoleic acid (CLA). They first prepared a GPC from PC derived from soybean and then used it to esterify with CLA under the action of Novozyme 435. Under the condition of 40 °C, 48 h, 1/50 substrate molar ratio GPC/CLA, 10% (*w/w*) enzyme load in a vacuum system, 70 mol% LPC was obtained.

During the production of SPLs by esterification, both GPL and LPL need to be prepared. LPL is the one-step hydrolysate of PLs, while GPL is the two-step hydrolysate. The preparation and separation of LPL/GPL may be much more complex than the esterification of them, so there have been few studies on it. Besides, esterification reaction is only possible and useful in a microaqueous reaction system where hydrolysis should be minimized by keeping limited amounts of catalytic and conformational water in the system [22]. Water, which is one of the direct products of the esterification reaction, has significant effects on the shifting of the reaction equilibrium. It has to be continuously removed from the system, increasing the complexity of the process.

**Figure 2.** Lipases catalyze the esterification between *sn*−glycerol−3−phospholipids (GPL) (**a**) or lys−phospholipids (LPL) (**b**,**c**) and free fatty acid (FFA) to produce corresponding SPLs.

### 2.2. Acidolysis

Acidolysis occurs between an ester and an acid, resulting in an exchange of acyl groups. Figure 3 shows the scheme of the acidolysis reaction. As shown in Table 1, acidolysis could be used to incorporate free fatty acids (FFA) into PLs, such as anisic (ANISA) and veratric (VERA) [25], citronellic acid [26], EPA and DHA [27], conjugated linoleic acid (CLA) [28] and some other fatty acids. Because the properties of SPLs depend directly on the types of fatty acid residues, the incorporation of extraneous fatty acids may lead to the added value of phospholipids [29].

**Figure 3.** Lipase−mediated acidolysis reaction. Acidolysis occurs between an ester and an acid, resulting in an exchange of acyl groups.

**Table 1.** Lipase−mediated of SPLs production by acidolysis.

| FFA Type | Lipase | Reaction Conditions | Yield (%) | Reference |
|---|---|---|---|---|
| DHA and EPA from four different fish oil sources | RML (Lipase from *Rhizomucor miehei*) | 40 °C, 24 h around, 1/3.6 around substrate molar ratio SPLs/ω-3 fatty acids, 20% (*w/w*) enzyme load in organic solvent (hexane) | 70% ω-3 fatty acids-SPLs | [27] |
| Conjugated linoleic acid (CLA) | RML (Lipase from *Rhizomucor miehei*) | 45 °C, 48 h, 1/6 substrate molar ratio PC/CLA, 24% (*w/w*) enzyme load in organic solvent (heptane and 5 *v/v*% DMF) | 50% incorporation of CLA isomers into PC | [30] |
| Conjugated linoleic acid (CLA) | RML (Lipase from *Rhizomucor miehei*) | 45 °C, 36 h, 1/8 substrate molar ratio PC/CLA, 24% (*w/w*) enzyme load in organic solvent (heptane) | 33.8% and 50.1% incorporation of CLA into PC and LPC, respectively | [28] |
| Conjugated linoleic acid (CLA) obtained from sunflower and safflower | RML (Lipase from *Rhizomucor miehei*) | 45 °C, 36 h, 1/8 substrate molar ratio PC/CLA, 24% (*w/w*) enzyme load in organic solvent (heptane) | 42% incorporation of CLA into PC | [31] |
| Anisic (ANISA) and veratric (VERA) | Novozyme 435 (Lipase B from *Candida Antarctica*) | 50 °C, 72 h, 1/15 substrate molar ratio PC/ANISA, 30% (*w/w*) enzyme load in the selected binary solvent system | 28.5% (*w/w*) ANISA-LPC and 2.5% (*w/w*) ANISA-PC | [25] |
| p-Methoxcinnamic acid (p-MCA) | Novozyme 435 (Lipase B from *Candida Antarctica*) | 50 °C, 72 h, 1/10 substrate molar ratio PC/Ep-MCA, 30% (*w/w*) enzyme load in a binary solvent system of toluene/chloroform 9/1 (*v/v*) | 32% (*w/w*) p-MCA-LPC and 3% (*w/w*) p-MCA-PC | [32] |
| Citronellic Acid | Novozyme 435 (Lipase B from *Candida Antarctica*) | 30 °C, 48 h, 1/60 substrate molar ratio PC/CA, 30% enzyme load in organic solvent (toluene) | 33% CA-PC | [26] |
| Caprylic acid | Lipozyme TL IM (Lipase from *Thermomyces lanuginose*) | 55 °C, 70 h, 1/6 substrate molar ratio PC/caprylic acid, 40% enzyme load in a solvent-free system | 46% incorporation of caprylic acid into PC | [33] |
| Caprylic acid | Lipozyme TL IM (Lipase from *Thermomyces lanuginose*) | 54 °C, 50 h, 1/15 substrate molar ratio PC/caprylic acid, 29% enzyme load in organic solvent (hexane) | 46% incorporation of caprylic acid into PC | [34] |
| Caprylic acid | Lipozyme TL IM (Lipase from *Thermomyces lanuginose*) | Continuous production in a packed bed reactor | 25% incorporation of caprylic acid into PC | [35] |
| Caprylic acid | Lipozyme TL IM (Lipase from *Thermomyces lanuginose*) | 57 °C, 70 h, 1/5.5 substrate molar ratio PLs/caprylic acid, 30% (*w/w*) enzyme load in a solvent-free system | 39% incorporation of caprylic acid into PLs | [36] |

RML (lipase from *Rhizomucor miehei*), Novozyme 435 (lipase B from *Candida Antarctica*), and Lipozyme TL IM (lipase from *Thermomyces lanuginose*) are usually used in the production of SPLs. Rychlicka, Niezgoda and Gliszczynska [26] incorporated citronellic acid into egg yolk PC under the action of Novozyme 435. The modified phospholipid fraction enriched with citronellic acid in the *sn*–1 position (39% of incorporation) was obtained at a high 33% yield under the conditions of 30 °C, 48 h, 1/60 substrate molar ratio PC/CA, and 30% enzyme load in organic solvent (toluene). In 2020, Okulus and his group also used Novozyme 435 as the biocatalyst to catalyze the incorporation of anisic acid (ANISA) into PC [25]. Moreover, they indicated that Novozyme 435 was considered a non-regiospecific lipase, but in the majority of lipid modifications, it showed high selectivity toward the *sn*–1 position of PC. Hence, they presented the possible pathway of the reaction as shown in Figure 4. As for other lipases, Estiasih, Marianty and Ahmadi [27] used RML to catalyze the incorporation of DHA and EPA from four different fish oil sources into palm pressed fiber PLs. Vikbjerg, Peng, Mu and Xu [35] and Peng, Xu, Mu, Hoy and Adler-Nissen [36] both used Lipozyme TL IM to catalyze the incorporation of caprylic acid into soy PLs.

**Figure 4.** The possible pathway of the enzymatic synthesis of anisolylated phospholipids during the acidolysis reaction of egg−yolk phosphatidylcholine (PC) with anisic acid (ANISA). Reprinted with permission from Ref. [25]. 2020, MDPI.

The acidolysis reaction is reversible as well and is commonly considered a two-step reaction: hydrolysis and esterification. Generally, free fatty acids are chemically unstable, so such acidolysis reactions are usually conducted under mild conditions. Compared with the esterification strategy, there is no need to prepare LPC or GPC before the acidolysis. It is an advantage that only one process step is needed. However, the first hydrolysis reaction varies in degree, which may cause the mixture of fatty acids-LPC and fatty acids-PC as shown in Figure 4. To increase the purity of the major product, the appropriate reaction conditions including enzyme, temperature, time, substrate molar ratio, the enzyme load, solvent, and some other parameters must be carefully thought through.

### 2.3. Transesterification

Transesterification occurs between two esters, resulting in an exchange of acyl groups. Figure 5 shows the scheme of the transesterification reaction. Compared with acidolysis, the substrates of transesterification are esters rather than an ester and a free fatty acid, which can help stabilize the pH and avoid the side effect of fatty acids on the system. Both hydrolysis and esterification occur simultaneously in the transesterification reaction, whereas the fatty

acid residues in the *sn*–1 or *sn*–2 position, which are dependent on the specificity of the lipase [37], are gradually exchanged with the added fatty acid residues in the forms of triacylglycerol (TAG), methyl ester or ethyl ester until equilibrium is reached [1].

**Figure 5.** Lipase−catalyzed transesterification for SPLs preparation. Transesterification occurs between two esters, resulting in an exchange of acyl groups.

The recent work on lipase-mediated SPLs production by transesterification is summarized in Table 2. In 2020, Rychlicka, et al. [38] synthesized structured *O*-methylated phenophospholipids by the transesterification reaction of egg-yolk PC with ethyl ester of 3,4-dimethoxycinnamic acid (E3,4DMCA) catalyzed by Novozyme 435 in hexane as a reaction medium. The optimization of this process was performed by statistical design methods and under the condition of 50 °C, 72 h, 1/10 substrate molar ratio PC/E3,4DMCA, and 30% (*w/w*) enzyme load, 27.5% (*w/w*) 3,4DMCA-LPC and 3.5% (*w/w*) 3,4DMCA-PC could be obtained. DHA and EPA can be incorporated into PLs by the transesterification reaction in the form of ethyl ester as well. Both Marsaoui, et al. [39] and Wang, et al. [40] have modified soy PC with n-3 polyunsaturated fatty acids (PUFA)-rich ethyl esters (EE) by immobilized lipase RML and MAS1, respectively. The high level of incorporation obtained (56.8% of Marsaoui's group) suggests that enzymatic production of "marine phospholipid from soybean lecithin" for the food and nutraceutical industries holds promise [39]. Besides, α-linolenic acid (ALA) [41], punicic acid (PA) [42], oleic acid [43], and myristic acid (MA) [44] can also be incorporated into PLs through transesterification.

**Table 2.** Lipase-mediated of SPLs production by transesterification.

| PL Source | Acyl Donor | Lipase | Reaction Conditions | Yield (%) | Reference |
|---|---|---|---|---|---|
| Palm pressed fiber PLs | DHA and EPA from four different fish oil sources | RML (Lipase from *Rhizomucor miehei*) | 40 °C, 24 h around, 1/3.6 around substrate molar ratio SPLs/ω-3 fatty acids, 20% (*w/w*) enzyme load in organic solvent (hexane) | 70% ω-3 fatty acids-SPLs | [27] |
| Egg yolk PC | Conjugated linoleic acid (CLA) | RML (Lipase from *Rhizomucor miehei*) | 45 °C, 48 h, 1/6 substrate molar ratio PC/CLA, 24% (*w/w*) enzyme load in organic solvent (heptane and 5 *v/v*% DMF) | 50% incorporation of CLA isomers into PC | [30] |
| Egg yolk PC | Conjugated linoleic acid (CLA) | RML (Lipase from *Rhizomucor miehei*) | 45 °C, 36 h, 1/8 substrate molar ratio PC/CLA, 24% (*w/w*) enzyme load in organic solvent (heptane) | 33.8% and 50.1% incorporation of CLA into PC and LPC, respectively | [28] |

**Table 2.** *Cont.*

| PL Source | Acyl Donor | Lipase | Reaction Conditions | Yield (%) | Reference |
|---|---|---|---|---|---|
| Egg yolk PC | Conjugated linoleic acid (CLA) obtained from sunflower and safflower | RML (Lipase from *Rhizomucor miehei*) | 45 °C, 36 h, 1/8 substrate molar ratio PC/CLA, 24% (*w/w*) enzyme load in organic solvent (heptane) | 42% incorporation of CLA into PC | [31] |
| Egg yolk PC | Anisic (ANISA) and veratric (VERA) | Novozyme 435 (Lipase B from *Candida Antarctica*) | 50 °C, 72 h, 1/15 substrate molar ratio PC/ANISA, 30% (*w/w*) enzyme load in the selected binary solvent system | 28.5% (*v/v*) ANISA-LPC and 2.5% (*v/v*) ANISA-PC | [25] |
| Egg yolk PC | p-Methoxcinnamic acid (p-MCA) | Novozyme 435 (Lipase B from *Candida Antarctica*) | 50 °C, 72 h, 1/10 substrate molar ratio PC/Ep-MCA, 30% (*w/w*) enzyme load in a binary solvent system of toluene/chloroform 9/1 (*v/v*) | 32% (*w/w*) p-MCA-LPC and 3% (*w/w*) p-MCA-PC | [32] |
| Egg yolk PC | Citronellic Acid | Novozyme 435 (Lipase B from *Candida Antarctica*) | 30 °C, 48 h, 1/60 substrate molar ratio PC/CA, 30% enzyme load in organic solvent (toluene) | 33% CA-PC | [26] |
| Soy PC | Caprylic acid | Lipozyme TL IM (Lipase from *Thermomyces lanuginose*) | 55 °C, 70 h, 1/6 substrate molar ratio PC/caprylic acid, 40% enzyme load in a solvent-free system | 46% incorporation of caprylic acid into PC | [33] |
| Soy PC | Caprylic acid | Lipozyme TL IM (Lipase from *Thermomyces lanuginose*) | 54 °C, 50 h, 1/15 substrate molar ratio PC/caprylic acid, 29% enzyme load in organic solvent (hexane) | 46% incorporation of caprylic acid into PC | [34] |
| Soy PC | Caprylic acid | Lipozyme TL IM (Lipase from *Thermomyces lanuginose*) | Continuous production in a packed bed reactor | 25% around the incorporation of caprylic acid into PC | [35] |
| Soy PLs | Caprylic acid | Lipozyme TL IM (Lipase from *Thermomyces lanuginose*) | 57 °C, 70 h, 1/5.5 substrate molar ratio PLs/caprylic acid, 30% (*w/w*) enzyme load in a solvent-free system | 39% incorporation of caprylic acid into PLs | [36] |

One of the shortcomings of transesterification, however, is that the fatty acid composition in the *sn*–1 or *sn*–2 position of the product will be a mixture of the original fatty acid and the one to be incorporated. To increase the purity of the major product, a high excess of the fatty acyl to be incorporated must be used [45]. Moreover, the reaction mechanism of some novel immobilized lipase-catalyzed transesterification of fatty acyl and PLs is still unknown, which should be studied further to know how to control the hydrolysis reaction during the process.

## 3. Phospholipase-Catalyzed Structured Phospholipids

Phospholipases can catalyze the production of SPLs as well by hydrolyzing PLs at different ester bonds in terms of their specificities. Generally, phospholipases could be grouped into four classes, namely A, B, C, and D [22]. As shown in Figure 6, phospholipase A (PLA) mainly cleaves the ester bond at either the *sn*–1 position (phospholipase $A_1$, $PLA_1$) or the *sn*–2 position (phospholipase $A_2$, $PLA_2$). Phospholipase B (PLB) hydrolyzes the ester bond at both the *sn*–1 position and *sn*–2 position. Phospholipase C (PLC) cleaves the phosphodiester bond at the glycerol backbone, while phospholipase D (PLD) removes the polar head group (X) of PLs. Previous studies have demonstrated that phospholipases play crucial roles in cellular regulation, metabolism, biosynthesis, and selective modification of PLs [16]. Despite the catalytic potential of phospholipases in the production of SPLs, the low productivity hinders their practical applications [5]. To increase the production and enhance the catalytic performance, genetic engineering and protein engineering on phospholipases have been put into effect [46,47]. Zhang, et al. [48] designed a novel two-step expression

system to produce and secrete recombinant PLD in the extracellular medium, and cellulose-binding domains as an affinity fused with PLD for immobilization and purification proteins, demonstrating great potential in industrial applications. Damnjanovic, et al. [49] altered the *sn*−2 acyl chain recognition of a PLD, leading to a variant enzyme preferably reacting on lysophospholipids (LPL) by protein engineering, to discriminate between PLs and LPL.

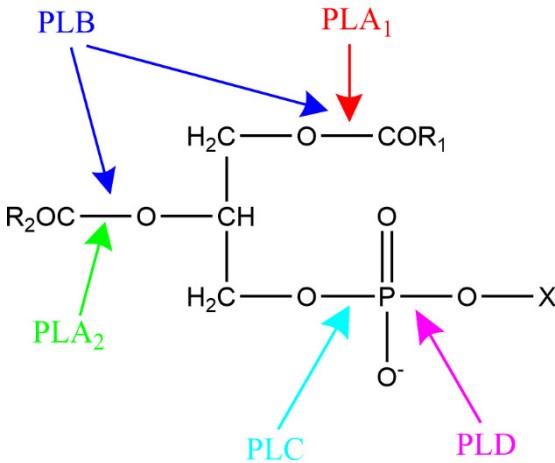

**Figure 6.** The acting sites of phospholipases. Phospholipase A (PLA) cleaves the ester bond at either the *sn*−1 position (phospholipase A$_1$, PLA$_1$) or the *sn*−2 position (phospholipase A$_2$, PLA$_2$). Phospholipase B (PLB) hydrolyzes the ester bond at both the *sn*−1 position and *sn*−2 position. Phospholipase C (PLC) cleaves the phosphodiester bond at the glycerol backbone, while phospholipase D (PLD) removes the polar head group (X) of PLs.

Compared with lipases, phospholipase A, and phospholipase D, there are fewer studies on phospholipase B- or C-mediated SPLs. As for phospholipase B (EC 3.1.1.5), it can catalyze the hydrolytic cleavage of fatty acids that are esterified both at the *sn*–1 and *sn*–2 positions. The hydrolysis reaction produces FFAs and LPC or GPC. To date, the structural information and catalytic mechanism of most PLBs have not been clear [50], hindering the application of PLB in the production of SPLs. As for phospholipase C (EC 3.1.4.11), it hydrolyzes the phospholipids on the diacylglycerol side of the phosphodiester bond producing phosphomonoesters and diacylglycerols (DAG) [51]. Under the action of PLC, phospholipids will be transferred into the glyceride form, so PLC is not suitable for the production of SPLs.

### 3.1. Phospholipase A-Mediated the Preparation of Structured Phospholipid

According to different acting sites, PLA can be further classified into PLA$_1$ and PLA$_2$. Table 3 lists recent literature on PLA-mediated of SPLs production.

**Table 3.** PLA-mediated SPLs production.

| PL Source | Acyl Donor | Enzyme | Reaction Conditions | Yield (%) | Reference |
|---|---|---|---|---|---|
| Soy PC | - | PLA$_1$ | 60 °C, 9 h, 200 U/g enzyme load in water system | over 100% hydrolysis rate | [52] |
| Soy PC | Conjugated linolenic acid (CLA) | PLA$_1$ immobilized on Duolite A658 | 50 °C, at least 24 h, 1/4 substrate molar ratio of PC/CLA, and 15% (*w/w*) enzyme load | Maximal (>99%) molar yield of structured PC with 72.3% CLA content | [53] |
| Antarctic krill PLs | EPA and DHA | PLA$_1$ | 55.22 °C, 24 h, 1/5.15 substrate molar ratio of PLs/FFA, 20% (*w/w*) enzyme load in organic solvent (hexane) | 64.35% incorporation of EPA/DHA into PLs | [54] |

**Table 3.** *Cont.*

| PL Source | Acyl Donor | Enzyme | Reaction Conditions | Yield (%) | Reference |
|---|---|---|---|---|---|
| Soy PC | DHA-enriched fatty acids | PLA$_1$ | 45 °C, 24 h, 1/2.13 substrate molar ratio of PC/fatty acids, 40% (*w/w*) enzyme load in the selected binary solvent system | 20.90% incorporation of DHA into PC | [55] |
| Soy PC | Free medium-chain fatty acid | PLA$_1$ immobilized on Duolite A568 | 45 °C, 24 h, 1/15 substrate molar ratio of PC/free medium-chain fatty acid, 12% (*w/w*) enzyme load in solvent-free system | 52.98% modified PC | [56] |
| Soy PC | DHA/EPA-rich ethyl esters | PLA$_1$ immobilized on macroporous resin | 55.7 °C, 24 h, 1/6.8 substrate molar ratio of PC/ethyl ester, and 15% (*w/w*) enzyme load | 19.09% incorporation of DPA/EPA into PC | [57] |
| Soy PC | Capric acid | PLA$_1$ immobilized on Amberlite XAD 7HP | 45 °C, 72 h, 1/10 substrate molar ratio of PC/capric acid, 10% (*w/w*) enzyme load in organic solvent (hexane) | 51.0 mol% incorporation of capric acid into PC | [58] |
| Soy PC | Fatty acid enriched with n-3 PUFA from fish oil | PLA$_1$ immobilized on VP OC 1600 | 55 °C, 24 h, 1/8 substrate molar ratio of PC/fatty acids, 20% (*w/w*) enzyme load in a solvent-free system | 57.4 mol% incorporation of n-3 PUFA into PC | [59] |
| Soy PC | DHA/EPA-rich ethyl esters | PLA$_1$ immobilized on resin D380 | 55 °C, 24 h, 1/6 substrate molar ratio of PC/ethyl esters, 15% (*w/w*) enzyme load in solvent-free system | 16.5% PC, 26.3% 1-LPC, 31.4% 2-LPC and 25.8% GPC | [56,60] |
| Soy PC | Medium-chain fatty acids (MCFAs) | PLA$_1$ immobilized on Duolite A568 | 50 °C, 72 h, 1/16 substrate molar ratio of PC/MCFAs, 16% (*w/w*) enzyme load in solvent-free system | 41% incorporation of free MCFAs into PC | [61] |
| Soy PC | Conjugated linolenic acid (CLA) | PLA$_1$ immobilized on Duolite A568 | 50 °C, 24 h, 1/4 substrate molar ratio of PC/CLA, and 15% (*w/w*) enzyme load | 90% incorporation of CLA into PC | [62] |
| Granulated PC | Free fatty acids enriched in EPA and DHA from a fish oil concentrate | PLA$_1$ from *Thermomyces lanuginosus/Fusarium oxysporum* immobilized on Duolite A568 | 50 °C, 48 h, 1/8 substrate molar ratio of PC/FFA, 10% (*w/w*) enzyme load in solvent-free system | 50 mol% around n-3 PUFA content of total LPC residues | [63] |
| Soy PC | Concentrated fish oil enriched in n-3 fatty acids | PLA$_1$ immobilized on Duolite A568 | 50 °C, 12 h, 1/8 substrate molar ratio of PC/free fatty acids, 10% (*w/w*) enzyme load in solvent-free system | Nearly 35% of the total esterified fatty acid residues were n-3 species (EPA, DPA, or DHA) | [64] |
| Soy PC | Medium-chain fatty acids (MCFAs) | PLA$_2$ | 40 °C, 70 h, 1/6 substrate molar ratio of PC/caprylic acid, 40% (*w/w*) enzyme load in organic solvent (hexane) | 87.7% incorporation of caprylic acid into PC | [65] |
| Soy PC | Caprylic acid | PLA$_2$ immobilized on the hydrophobic resin Diaion HP-20 | 50 °C, 48 h, 1/12 substrate molar ratio of PC/caprylic acid, 50% (*w/w*) enzyme load in organic solvent (hexane) | 45.29 mol% the ML-type PC (M: medium-chain fatty acid; L: long-chain fatty acid) | [66] |
| Epikuron 200 (PC, 93%) | Caprylic acid | PLA$_2$ immobilized on Amberlite XAD7 | 45 °C, 48 h, 1/9 substrate molar ratio of PC/caprylic acid, 30% (*w/w*) enzyme load in solvent-free system | 36% incorporation of caprylic acid | [37] |

PLA$_1$ (EC 3.1.1.32) is also a member of the triacylglycerol lipase family and shows considerable sequence similarity to the guinea pig pancreatic lipase-related protein 2 (GPLRP2) and human hepatic and pancreatic lipases [22]. Many lipases share amino-acid sequences identical to PLA$_1$ and show phospholipase activity [5]. As shown in Table 3, PLA$_1$ displayed greater catalytic performance in the incorporation of EPA/DHA [54,60,63], medium-chain fatty acids (MCFAs) [56,61], conjugated linolenic acid (CLA) [62,67] and some other fatty acids into PLs via the acidolysis or transesterification reaction (Figure 7).

Immobilized PLA$_1$-catalyzed transesterification of PC and DHA/EPA-rich ethyl esters is similar to the lipase-catalyzed transesterification [57].

(a) Acidolysis

(b) Transesterification

**Figure 7.** PLA$_1$−catalyzed incorporation of different fatty acids into PLs via the acidolysis (**a**) or transesterification (**b**) reaction.

PLA$_2$ (EC 3.1.1.4) can catalyze the exchange of fatty acid in the *sn*–2 position via acidolysis or the transesterification reaction [37,65] (Figure 8). Compared with a lipase- and PLA$_1$- catalyzed reaction, PLA$_2$-catalyzed acyl exchange has received little attention as PLA$_2$ may lack the potential to catalyze the transesterification reaction since no acyl-enzyme intermediate is formed [66].

(a) Acidolysis

(b) Transesterification

**Figure 8.** PLA$_2$−catalyzed the exchange of fatty acid in the *sn*−2 position via acidolysis (**a**) or transesterification (**b**) reaction.

Most commercial products of PLAs are provided in the liquid solution (free form). To economize the process, PLAs are usually immobilized before being used in catalysis. Resinous materials, such as Duolite A658 [56,62], resin D380 [60], Amberlite XAD 7HP [37,58] and some other supports are the most commonly used and appropriate choices to immobilize PLA. Both the properties of supports (such as pore size, stability, hydrophobicity, electronegativity, etc.) and the conditions of the immobilization (such as temperature, pH, time, the amount of the enzyme, etc.) determine the immobilization efficiency and

catalytic performance of the immobilized enzymes. Besides immobilized enzymes, using reverse micelles as the reaction system became an alternative to avoid the process of immobilization and to ensure the efficient catalysis of PLA [55].

### 3.2. Phospholipase D-Mediated the Preparation of Structured Phospholipid

PLD (EC.3.1.4.4) can catalyze both the hydrolysis of PLs to phosphatidic acid (PA) and the transphosphatidylation of PC to PLs when appropriate acceptor alcohols are provided, as shown in Figure 9. Compared with lipase- and PLA-mediated of SPLs production, PLD-mediated production mainly aims to exchange, or remove the polar headgroup on the *sn*–3 position of PLs rather than fatty acid residues on the *sn*–1 or *sn*–2 positions. PLDs are now recognized as important tools for the enzymatic synthesis of SPLs, and Table 4 lists recent literature on PLD-mediated of SPLs production.

**Figure 9.** PLD−mediated transphosphatidylation to prepare phosphatidyl serine (**e**), phosphatidylinositol (**f**), phosphatidyl glycerol (**g**), and phosphatidyl ethanolamine (**h**) from PC and L−serine (**a**), myo−inositol (**b**), glycerol (**c**), and ethanolamine (**d**).

**Table 4.** PLD-mediated of SPLs production.

| PL Source | Acceptor Alcohol | Enzyme | Reaction Conditions | Yield (%) | Reference |
|---|---|---|---|---|---|
| Soy PC | L-serine | PLD from *marine Streptomyces klenkii* (SkPLD) | 40 °C, 12 h, 1/5 substrate mass ratio of PC/L-serine, 25% (*w/w*) enzyme load in organic solvent (hexane) | 26.18% yield of PS | [68] |
| Soy PC | L-serine | PLD from *Streptomyces* sp. | 30 °C, 70 min, alcohol to PL ratio of 100/1, and 25% (*v/w*) enzyme load | 98.3% yield of PS | [69] |
| Soy PC | Tyrosol | PLD from *Actinamadure* sp. | 30 °C, 8 h, 1/20 substrate mass ratio of PC/Tyrosol, 1% (*w/w*) enzyme load in the selected binary solvent system | 94% PC conversion | [70] |
| Soy PC | Glucose | PLD from *Streptomyces* sp. | 60 °C, 1.5 h, 1/20 substrate mass ratio of PC/Glucose, 30 U/mL enzyme load in the biphasic reaction system | 95 mol% PL-Glu | [71] |
| Soy PC | Phenylalkanols | PLD from *Streptomyces* sp. | 37 °C, 24 h, 1/10 substrate mass ratio of PC/phenylalkanols, 1.6 U enzyme in the biphasic reaction system | 87% yield of phosphatidyl-tyrosol | [72] |

To date, several microorganisms have been claimed to have the ability to produce PLDs, containing Streptomyces sp., Bacillus cereus, Escherichia coli, Acinetobacter radioresistance, Ochrobactrum sp., Pseudomonas aeruginosa, Salmonella typhimurium, and Corynebacterium sp. [16]. Among these sources, PLDs from Streptomyces sp. show the highest transphosphatidylation activity and broadest substrate specificity, thus becoming the most commonly used PLDs for the production of SPLs [46]. As shown in Table 4, serine, tyrosol, glucose, phenylalkanols, and some other organic compound with hydroxyl groups could be used as the acceptor alcohol and further synthesize the corresponding SPLs. Casado, Reglero and Torres [70] used a food-grade phospholipase D from Actinamadure sp. to produce highly purified phosphatidyl-tyrosol in a GRAS (Generally Recognized as Safe) biphasic medium. Figure 10 shows the relative reaction scheme. Then the procedure was scaled up using 40 g of highly purified phosphatidyl-tyrosol (97 wt%) without involving the utilization of organic solvents, which provided a strategy that may meet the need of food industrial applications.

**Figure 10.** Schematic presentation of phospholipase D (PLD) transphosphatidylation reaction. Casado, Reglero and Torres used a food−grade phospholipase D from *Actinamadure* sp. to produce highly purified phosphatidyl−tyrosol in a GRAS (Generally Recognized as Safe) biphasic medium. Reprinted with permission from Ref. [70]. 2013, Elsevier.

To expand the application of PLDs, searching for novel PLDs with a higher transphosphatidylation activity, analyzing the crystal of more PLDs from different sources, and improving the catalytic performance of PLDs through rational design should be paid more attention to in the future [16]. Besides, with the improvement of the requirements of safety and health, food-grade expression systems and a hosts for PLDs are in need.

## 4. Factors Influencing Enzyme-Mediated Structured Phospholipids and Related Mechanism

Several factors account for a lot in the process of enzymatic producing SPLs. To optimize the reaction conditions, response surface methodology (RSM) is the most common and effective statistical technique used in various research. Via the experiment design, the significant parameters could be identified and optimized in a specific system. Though the optimal conditions may vary on different occasions, some parameters that commonly affect the SPLs production and related mechanism could be summarized as follows.

### 4.1. Reaction Medium/Solvent

Whether solvent exists or not determines the production of SPLs. The presence of solvent would improve mixing in the system and make the subsequent removal of the enzymes more convenient in favor of the industry practice [73]. Many studies showed that the amount of solvent had a very significant effect on the yield [34,74]. Vikbjerg, Mu and Xu [33] pointed out, that with an increasing amount of hexane, the recovery of PC decreased, probably as a result of increased hydrolysis. Since the amount of solvent reduced

the recovery of PC more strongly than it increased the incorporation, it is recommended that this should be kept as low as possible. At the same time, the use of solvents would increase the capital investment when the process is scaled up. Preferably the reaction should be conducted in solvent-free systems.

The polarity of the reaction medium influences the reaction of SPLs production as well. The polarity would affect the contact among enzymes, PLs and fatty acid molecules, as well as have an impact on the acyl migration of SPLs. In addition to the solvent, using PLs with different polar headgroups or adding different ratios of substrates (PLs/FFA) could also change the polarity of the reaction medium [1].

### *4.2. Acyl Migration*

It is proved that an n-3 PUFA at the *sn*–2 position benefits health and has potential applications in the food industry [36,74]. However, during the enzyme-mediated production of SPLs, non-enzymatic acyl migration from the *sn*–2 position to the *sn*–1 position or the opposite direction is harmful to the purity and function of the desired products.

Figure 11 shows the mechanism of the acyl migration from the *sn*–2 position to the *sn*–1 position under acid (a) and base (b) conditions, respectively. In the process of acyl migration, the carbonium ion of fatty acid acyl is attacked by the lone pairs of electrons of the hydroxyl oxygen atom at the *sn*–1 position and then forms a five-membered ring intermediate (**m** and **n** in Figure 11). Then the intermediate fractures to form a more stable compound, *sn*–1-LPC. The formation of the five-membered ring is the rate-limiting step of the acyl migration. The process is influenced by complex parameters containing the solvent, temperature, enzyme, and some other conditions.

**Figure 11.** The mechanism of the acyl migration from the *sn*−2 position to the *sn*−1 position under acid (**a**) and base (**b**) conditions. In the process of acyl migration, the carbonium ion of fatty acid acyl is attacked by the lone pairs of electrons of the hydroxyl oxygen atom at the *sn*−1 position and then forms a five−membered ring intermediate (**m** and **n**). Then the intermediate fractures to form a more stable compound, *sn*−1−LPC.

Increasing solvent polarity or the addition of water to nonpolar solvents has been reported to cause lower rates of acyl migration [74]. However, Zhang, et al. [75] found that *sn*–2 LPC had poor solubility in low polarity solvents, such as hexane, isooctane, and ethyl acetate; it could dissolve better in high polarity solvents containing water, methanol, and isopropanol, increasing the rate of acyl migration. Zhang, Zhang, Qu, Wang and Liu [75] also studied the effect of pH on acyl migration. They pointed out that both the acid and base conditions would deepen the degree of acyl migration. With the increase

in the concentration of the hydrogen or hydroxide ions, the possibility and rate of the five-membered ring formation increase, thus improving the acyl migration.

The temperature is also reported to affect acyl migration. According to Arrhenius's law, an increase in the reaction temperature of the enzyme-catalyzed reactions resulted in increased reaction rates. Moreover, for endothermic reactions, a higher temperature is beneficial to obtain a higher yield due to a shift in the thermodynamic equilibrium [55]. The higher the temperature is, the higher the rate of molecular velocity is, adding to the possibility of the formation of the five-membered ring. Thus, with the increase in temperature, the acyl migration in the reaction system increases [74,76]. In another word, excessive temperature increases the occurrence of side reactions and decreases the purity of the major products.

Both the supports and the dosage of enzymes affect acyl migration. Various carriers, including silica, diatomite, ion exchange resin, and adsorption resin, have different effects on acyl migration. Some supports, such as resin and silica are potential catalysts for acyl migration. They can promote the hydroxyl protonation of the acyl receptor, thus increasing nucleophilicity and accelerating the migration of the acyl group [77]. In the reactions which used RML immobilized on the anion exchange resin, and Lipozyme RM IL (commercial lipases immobilized on resin), and Lipozyme TL IM (commercial lipases immobilized on silica) as catalysts, the acyl migration was observed, while for *Rhizopus oryzae* lipase immobilized on polypropylene, no acyl migration was observed in the reaction [78,79]. Meanwhile, Vikbjerg, Mu and Xu [74] found that the enzyme load also had a significant effect on acyl migration. With the increase in the enzyme dosage, the degree of acyl migration increased. That is because a high enzyme dosage would carry more water into the reaction medium, increasing the hydrolysis of PLs and promoting the production of LPC, thus raising the possibility of acyl migration.

In addition, the concentration of PLs was reported to affect the acyl migration. As *sn*–2 LPC has a hydrophilic "head" and a hydrophobic "tail", at high concentrations, PL molecules would aggregate into small liposomes, inhibiting the stretch of the fatty acid acyl and decreasing the acyl migration. At low concentrations, however, the rate of acyl migration also decreased because the distance between two *sn*–2 LPC is larger, which reduces the synergistic interactions between molecules. Zhang, Zhang, Qu, Wang and Liu [75] found that the acyl migration rate would increase with the increase in the concentration of *sn*–2-LPC from 0.5 to 5 mg/mL, while the migration rate would decrease when the concentration increased from 5 to 20 mg/mL. The maximum rate of acyl migration could reach 32% when the concentration of *sn*–2-LPC was 5 mg/mL. It was also reported that the acyl migration could be suppressed with the increase in the degree of unsaturation of PLs because the nucleophilicity of the carbonium ion decreased. Lim, et al. [80] found the pattern and optimized the reaction conditions to minimize the undesired acyl migration. Finally, 83.7% LPC was obtained.

### 4.3. Water Content/Activity

Water plays a vital and complex role in the production of SPLs. On one hand, the presence of water in the reaction system is crucial for the hydrolysis of PLs and the activation of the enzyme. On another hand, under high content or activity of water, the additional moisture may either decrease the catalytic of the enzyme or lead to undesired hydrolysis reactions [30,63].

Optimal water addition is needed for SPLs production. Wang, et al. [81] claimed that the incorporation of n-3PUFA into PC increased with increasing the water dosage from 0.5% to 1.0%. The maximum incorporation of n-3 PUFA (33.49%) was observed at a water dosage of 1.0%, while the incorporation decreased when the water dosage was further increased from 1.0% to 1.5%. This is probably because that excess water resulted in the undesired hydrolysis reactions. Li, Qin, Wang, Li, Yang and Wang [57] found that water addition was the most significant factor in determining the incorporation of DHA/EPA into PC under the action of immobilized $PLA_1$. The incorporation of EPA/DHA was

higher at a water addition of 1.1 wt%, than that at 0, 0.5, 0.75, and 1.25 wt%. Levels of water addition also affected reaction time where the incorporation reached its maximum under a given water dosage and the final composition of the products. Niezgoda and Gliszczynska [30] investigated the effect of water activity on the efficiency of PC acidolysis with conjugated linoleic acid (CLA) under the condition of $a_w$ = 0.11, 0.23, and 0.33. They found that the effective incorporation of CLA declined in the series 0.23 > 0.33 > 0.11. Lowering $a_w$ from 0.33 to 0.23 resulted in the more effective production of SPLs, while a very low content of water would inhibit the hydrolysis reaction and destroy the basic conformation of the enzyme.

*4.4. Oher Conditions*

As another leading role in the production of SPLs, the ratio and reactivity of substrates also have a significant effect. As for substrate ratio, before the optimal FAs concentration, the increase in the ratio of the substrate may increase the yield for both acidolysis and transesterification [22,55]. The increase in the FAs concentration leads to the solubility of PLs, thus enhancing the combination between FAs and PLs. After the optimal ratio, however, the content of SPLs may decline. This is due to the increasing polarity and viscosity of the reaction system with the addition of FAs, which could hinder the spreading of substrates and reduce the opportunity for molecule collisions among FFA, PLs, and enzymes [61].

The reactivity of fatty acid substrates as acyl donors varies for different types. Egger, et al. [82] investigated different fatty acids used to incorporate into PC and reported that reaction rates are associated with the chain length and unsaturation degree of the fatty acids. The reaction rate decreased sharply when the length of FAs increased from C6 to C12, C14 (myristic acid), and C16 (palmitic acid) due to the lower solubility of these FAs. Oleic acid (C18:1) exhibited the highest reaction rate, while FAs with a higher degree of unsaturation (C18:2 and C18:3) were associated with lower reaction rates. The purity of the substrates also affects the activity of the reaction. By using purer conjugated linoleic acid (CLA) substrates, Peng, Xu, Mu, Hoy and Adler-Nissen [36] found that higher incorporation of CLA into PC could be obtained. Besides, the reactivity of PLs determines the production of SPLs. Peng, Xu, Mu, Hoy and Adler-Nissen [36] also investigated the acidolysis reaction with pure individual PL species and FAs. They found that PC showed the highest overall incorporation, followed by PE, PA, and PI. The results prove that the initial composition of PLs would affect the incorporation of FAs into PLs. Purer PC can lead to higher production, but for commercial considerations, soy PLs mainly containing PC might be favored since its price is considerably lower than pure PC.

## 5. Prospect

It is proved that structured phospholipids have high value in the field of pharmacy, food, cosmetics, and health. In recent years, plenty of research on enzyme-mediated SPLs was conducted. However, there are still some problems existing in the enzymatic production of SPLs, such as high cost of the procedure, low conversion of substrates, low productivity of major products, huge difficulty in expanding the production scale, and some other trouble. To achieve the practical application of enzyme-mediated SPLs as early as possible, more and more efforts should be put into the study of the modification of PLs.

First, hydrolysis, which is an unavoidable reaction during the formation of SPLs, should be further studied and controlled. More effective methods except for choosing appropriate water content/activity, reaction temperature and time, and using optimal lipases or phospholipases, should be attempted to inhibit the undesired hydrolysis of substrates and increase the purity of the target product in the future. Second, more attention needs to be paid to the study of the mechanism and regulation of the acyl migration, because the desired acyl migration would decrease the productivity and purity of the major products. Polarity, temperature, enzyme, PLs, and some other conditions have been proved to have an influence, but a deeper understanding of the mechanism and more

effective methods to regulate it are lacking. Since both the type and amount of the enzymes and PLs molecules would affect the acyl migration, it is crucial to choose or design a more rational synthesis scheme. Third, with the requirement of practical application, relative research on immobilized carrier engineering needs to be further conducted. The properties of supports containing pore size, stability, hydrophobicity, electronegativity, and some other characteristics determine the immobilization efficiency and catalytic performance of the immobilized enzymes. Hence, it is worth researching, designing and utilizing more novel and appropriate carriers to immobilize enzymes, increasing the efficiency of SPLs production.

**Author Contributions:** Y.L.: Literature collection and manuscript writing; L.D.: Literature collection and analysis; D.L.: Content design; W.D.: Content design, revision, project administration and funding acquisition. All authors have read and agreed to the published version of the manuscript.

**Funding:** This research was from National Natural Science Foundation of China (22178198).

**Data Availability Statement:** Not applicable.

**Conflicts of Interest:** The authors declare no conflict of interest.

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
