# Peer review of "Progress & Prospect of Enzyme-Mediated Structured Phospholipids Preparation"

_catalysts, doi:10.3390/catal12070795_

Round 1

Reviewer 1 Report

In this manuscript, the authors presented the current state of the art in enzyme-mediated structured phospholipids preparation. I have some suggestions. 

1. The authors mentionated that lipases and phospholipases are two main kinds of enzymes used to catalyze the production of SPLs, and they are widely found in animals, plants and microorganisms. This statement lacks references. On the other hand, the authors should give examples of animals, plants or microorganisms evidencing the abundance of these enzymes.

2. In addition to the titles, a brief description of the figures should be added in the legend.

3. To increase the production and enhance the catalytic performance, genetic engineering and protein engineering on phospholipases have been put into effect. Please provide examples to illustrate this point and enrich the manuscript.

4. In figure 6 the arrows and the identification of phospholipases could be of different color for better understanding.

5. Snake venoms are rich in phospholipase A2. Would these also be useful for the practical application discussed in this manuscript?

6. Do any of the phospholipases discussed require cofactors for enzymatic action?

7. The quality of figure 8 should be improved. The font size must be increased.

8. Scientific names are always italicized. Please review the first bottom paragraph of Table 4.

9. Some paragraphs only have one sentence. These must be expanded or combined with others according to idea, sequence and logic. Review the paragraph below figure 10.

10. The article presents an excess of figures, some already extensively explored in other articles or books. I suggest that the authors focus on the main and novelty for the area.

11. How are the enzymes used for the modification of phospholipids currently obtained? What are the projections and limitations?

Author Response

Response to Reviewer 1:

Reviewer’s comment 1): The authors mentioned that lipases and phospholipases are two main kinds of enzymes used to catalyze the production of SPLs, and they are widely found in animals, plants and microorganisms. This statement lacks references. On the other hand, the authors should give examples of animals, plants or microorganisms evidencing the abundance of these enzymes.

Response: We added references to evidence the abundance of these enzymes, and revised the expression as follows: Lipases and phospholipase are two main kinds of enzymes used to catalyze the pro-duction of SPLs, and they are widely found in animals (specific cells that synthesize lipases, or other host animals present in the gastrointestinal tract in animals to digest fats and lipids), plants (seeds or grains, fruits, leaves, and some other organisms) and microorganisms (fungi, yeast, and bacteria) [16,19-21].

Reviewer’s comment 2): In addition to the titles, a brief description of the figures should be added in the legend.

Response: We added a brief description of the figures in the legend.

Reviewer’s comment 3): To increase the production and enhance the catalytic performance, genetic engineering and protein engineering on phospholipases have been put into effect. Please provide examples to illustrate this point and enrich the manuscript.

Response: We added two examples to illustrate this point as follows: To increase the production and enhance the catalytic performance, genetic engineering and protein engineering on phospholipases have been put into effect [46,47]. Zhang, et al. [48] designed a novel two-step expression system to produce and secrete recombinant PLD in extracellular medium, cellulose-binding domains as an affinity fused with PLD for immobilization and purification proteins, demonstrating great potential in the industrial application. Damnjanovic, et al. [49] altering sn-2 acyl chain recognition of a PLD, leading to a variant enzyme preferably reacting on lysophospholipids (LPL) by protein engineering, to discriminate between PLs and LPL.

Reviewer’s comment 4): In figure 6 the arrows and the identification of phospholipases could be of different color for better understanding.

Response: We changed the color of the arrows and identification of phospholipases as Figure 6 shows.

Figure 6. The acting sites of phospholipases. Phospholipase A (PLA) cleaves the ester bond at either the sn-1 position (phospholipase A1, PLA1) or the sn-2 position (phospholipase A2, PLA2). Phospholipase B (PLB) hydrolyzes the ester bond at both the sn-1 position and sn-2 position. Phospholipase C (PLC) cleaves the phosphodiester bond at the glycerol backbone, while phospholipase D (PLD) removes the polar head group (X) of PLs.

Reviewer’s comment 5): Snake venoms are rich in phospholipase A2. Would these also be useful for the practical application discussed in this manuscript?

Response: As discussed in the review, PLA2 cleaves the ester bond at the sn-2 position and can be used to produce structured phospholipids (SPLs), which have great potential for application in the field of pharmacy, food, cosmetics, and health. Snake venoms contains a lot of proteins containing PLA2, and Siigur and Siigur [1] claimed that venom of Vipera berus berus contains about 15 protein protein/peptide families. In the theory, if PLA2 can be thoroughly separated from other components of snake venoms and meet the security requirements of public, they would be useful for the SPLs production as well.

Reviewer’s comment 6): Do any of the phospholipases discussed require cofactors for enzymatic action?

Response: Actually, cofactors are not mentioned in most of research on phospholipases-mediated structured phospholipids (SPLs) preparation. During the process of SPLs preparation, cofactors are not necessary. But cofactors are proved to affect the performance of phospholipases. For instance, Mohtar, et al. [2] reported that the addition of cofactors like Ca2+ could help to increase the activity of PLA2. Besides, some novel cofactors which participate in the regulation of physiological and pathological processes are explored as well [3-5].

Reviewer’s comment 7): The quality of figure 8 should be improved. The font size must be increased.

Response: We reviewed figures and found that figure 7 and figure 8 had the same font size. Maybe you are referring to figure 9 and we changed the font size of figure 9 as follows.

Figure 9. PLD-mediated transphosphatidylation to prepare phosphatidyl serine(a), phosphatidylinositol (b), phosphatidyl glycerol (c), and phosphatidyl ethanolamine (d) from PC.

Reviewer’s comment 8): Scientific names are always italicized. Please review the first bottom paragraph of Table 4.

Response: We reviewed and revised the sentence as follows: To date, several microorganisms have been claimed to have the ability to produce PLDs, containing Streptomyces sp., Bacillus cereus, Escherichia coli, Acinetobacter radioresistance, Ochrobactrum sp., Pseudomonas aeruginosa, Salmonella typhimurium, and Corynebacterium sp. [16].

Reviewer’s comment 9): Some paragraphs only have one sentence. These must be expanded or combined with others according to idea, sequence and logic. Review the paragraph below figure 10.

Response: We reviewed the paragraph and expanded it as follows: To expand the application of PLDs, searching for novel PLDs with a higher transphosphatidylation activity, analyzing the crystal of more PLDs from different sources, and improving the catalytic performance of PLDs through rational design should be paid more attention and action in the future [16]. Besides, with the improvement of the requirements of safety and health, food-grade expression system and host of PLDs are in need.

Reviewer’s comment 10): The article presents an excess of figures, some already extensively explored in other articles or books. I suggest that the authors focus on the main and novelty for the area.

Response: Thanks for suggestion. From our perspectives, although some of figures in this review have been already explored in other articles or books, it may be easier for the readers who are new to the field to understand relative research on enzyme-mediated structured phospholipids preparation if we retain these figures. Besides, we think that the lack of the existing figures may influence the integrity of the article. Thus, we would like to keep these figures in this review.

Reviewer’s comment 11): How are the enzymes used for the modification of phospholipids currently obtained? What are the projections and limitations?

Response: Most of study used the commercial enzymes to modify the phospholipids. For lipases, RML (lipase from Rhizomucor miehei), Novozyme 435 (lipase B from Candida Antarctica), and Lipozyme TL IM (Lipase from Thermomyces lanuginose) are commonly used as discussed in the review. And Lecitase® Ultra is the most common commercial phospholipase A1 used for the modification of phospholipids. In addition, some studies obtained the enzymes with high performance by applying genetic engineering [6-8] and protein engineering [9,10]. With the development of molecular simulation, directed evolution, rational design, high-throughput screen and some other techniques, more and more novel enzymes with specific traits would be exploited. Besides, immobilized enzymes would make the enzymes more practical for the practice. In the meantime, commercial enzymes usually have relatively high prices and the modifications of enzymes are generally tedious, both resulting in the high costs of the SPLs preparation.

References

  1. Siigur, J.; Siigur, E. Biochemistry and toxicology of proteins and peptides purified from the venom of Vipera berus berus. Toxicon: X 2022, 15, 100131, doi:10.1016/j.toxcx.2022.100131.
  2. Mohtar, L.G.; Ledesma, A.E.; Disalvo, E.A.; Frias, M.A. Influence of carbonyl groups on the interaction of PLA 2 with lipid interphases. Colloid and Interface Science Communications 2020, 39, doi:10.1016/j.colcom.2020.100309.
  3. Pothlichet, J.; Meola, A.; Bugault, F.; Jeammet, L.; Savitt, A.G.; Ghebrehiwet, B.; Touqui, L.; Pouletty, P.; Fiore, F.; Sauvanet, A.; et al. Microbial Protein Binding to gC1qR Drives PLA2G1B-Induced CD4 T-Cell Anergy. Frontiers in Immunology 2022, 13, doi:10.3389/fimmu.2022.824746.
  4. Xu, M.; Zhu, B.; Cao, X.; Li, S.; Li, D.; Zhou, H.; Olkkonen, V.M.; Zhong, W.; Xu, J.; Yan, D. OSBP-Related Protein 5L Maintains Intracellular IP3/Ca2+ Signaling and Proliferation in T Cells by Facilitating PIP2 Hydrolysis. Journal of Immunology 2020, 204, 1134-1145, doi:10.4049/jimmunol.1900671.
  5. Sherman, K.E.; Rouster, S.D.; Meeds, H.; Tamargo, J.; Chen, J.; Ehman, R.; Baum, M. PNPLA3 Single Nucleotide Polymorphism Prevalence and Association with Liver Disease in a Diverse Cohort of Persons Living with HIV. Biology-Basel 2021, 10, doi:10.3390/biology10030242.
  6. Samantha, A.; Damnjanovic, J.; Iwasaki, Y.; Nakano, H.; Vrielink, A. Structures of an engineered phospholipase D with specificity for secondary alcohol transphosphatidylation: insights into plasticity of substrate binding and activation. Biochemical Journal 2021, 478, 1749-1767, doi:10.1042/bcj20210117.
  7. Zhang, H.Y.; Li, X.H.; Liu, Q.; Sun, J.A.; Secundo, F.; Mao, X.Z. Construction of a Super-Folder Fluorescent Protein-Guided Secretory Expression System for the Production of Phospholipase D in Bacillus subtilis. Journal of Agricultural and Food Chemistry 2021, 69, 6842-6849, doi:10.1021/acs.jafc.1c02089.
  8. Zhang, H.Y.; Chu, W.Q.; Sun, J.A.; Liu, Z.; Huang, W.C.; Xue, C.H.; Mao, X.Z. A novel autolysis system for extracellular production and direct immobilization of a phospholipase D fused with cellulose binding domain. Bmc Biotechnology 2019, 19, doi:10.1186/s12896-019-0519-5.
  9. Damnjanovic, J.; Nakano, H.; Iwasaki, Y. Acyl chain that matters: introducing sn-2 acyl chain preference to a phospholipase D by protein engineering. Protein Engineering Design & Selection 2019, 32, 1-11, doi:10.1093/protein/gzz019.
  10. Fang, X.; Wang, X.T.; Li, G.L.; Zeng, J.; Li, J.; Liu, J.W. SS-mPEG chemical modification of recombinant phospholipase C for enhanced thermal stability and catalytic efficiency. International Journal of Biological Macromolecules 2018, 111, 1032-1039, doi:10.1016/j.ijbiomac.2018.01.134.

Reviewer 2 Report

In recent years, structured phospholipids (SPLs), which are modified phospholipids (PLs) have attracted more attention due to their great potential for application in the field of pharmacy, food,  cosmetics, and  health.  The present review paper includes the following parts: 1. Introduction, 2. Lipase-catalyzed structured phospholipids, 2.1. Esterification, 2.2. Acidolysis, 2.3. Transesterification, 3. Phospholipase-catalyzed structured phospholipids, 3.1. Phospholipase A- mediated the preparation of structured phospholipid, 3.2. Phospholipase D- mediated the preparation of structured phospholipid, 4. Factors influencing enzyme-mediated structured phospholipids and related mechanism, 4.1. Reaction medium/solvent, 4.2. Acyl migration, 4.3. Water content/activity, 4.4. Oher conditions and 5. Prospect. Very detailed and valuable are the detailed data presented in Table 1. Lipase-mediated of SPLs production by acidolysis, Table 2. Lipase-mediated of SPLs production by transesterification., Table 3. PLA-mediated SPLs production and Table 4. PLD-mediated of SPLs production. The review is interesting, but there is need to made minor revision regarding the following remarks.

The abstract should be modified. Since this is review article there is need to point out the range of years that is covered as well what is the novelty of present review.

At the end of introduction part there is need to state the novelty of present review.  Are there any other reviews of this topic? What about years range that is covered with present review? In addition, just simple search on Google scholar is reveling review papers on phospholipids (e.g. application in drug delivery systems, functional constituents of membranes, others).  Those review papers should be included in the introduction part and commented in short with the emphasis to the novelty of present review with respect to already published reviews on phospholipids.

Author Response

Response to Reviewer 2:

Reviewer’s comment 1): The abstract should be modified. Since this is review article there is need to point out the range of years that is covered as well what is the novelty of present review.

Response: We modified the abstract and the last sentence is expanded as follows: In this paper, the progress in enzymatic modification of PLs over last 20 years is reviewed. Reaction types and characteristic parameters are summarized in detail and the parameters affecting acyl migration are first discussed to give an inspiration to optimize the enzyme-mediated SPLs preparation. To expand the application of enzyme-mediated SPLs in the future, the prospect of further study on SPLs is also proposed at the end of the paper.

Reviewer’s comment 2): At the end of introduction part there is need to state the novelty of present review. Are there any other reviews of this topic? What about years range that is covered with present review? In addition, just simple search on Google scholar is reveling review papers on phospholipids (e.g. application in drug delivery systems, functional constituents of membranes, others). Those review papers should be included in the introduction part and commented in short with the emphasis to the novelty of present review with respect to already published reviews on phospholipids.

Response: We expanded the last paragraph of introduction part as follows: In recent years, several reviews on the SPLs have been published. Sun, Chen, Wang and Lin [2] presented a detailed review on the sources, molecular species, and structures of food-derived phospholipids. Wang, et al. [20] summarized developments in the formation of versatile phospholipid assemblies, together with the applications of these assemblies in building artificial tissues and biomedical applications. Ang, Chen, Xiang, Wei and Quek [1] provided information on the preparation method of SPLs, addressing potential health benefits and the methods used to analyze SPL profile and metabolism. This review, however, mainly focuses on the progress of enzyme-mediated structured phospholipids preparation, including the types of enzymes that are usually used in the production of SPLs with their corresponding strategies of reactions, and parameters affecting the SPLs production with related mechanisms. The progress in enzymatic modification of PLs over last 20 years is reviewed. To provide ideas for optimizing the enzyme-mediated SPLs preparation, reaction types and characteristic parameters are summarized in detail and the parameters affecting acyl migration are first discussed in this paper. Finally, the prospect of further study on SPLs is proposed to expand the application of enzyme-mediated SPLs in the future.

Round 2

Reviewer 1 Report

The authors presented substantial answers to my queries. In short lines, I find this manuscript ready for publication.